# Structural basis for amino acid transport by the CAT family of SLC7 transporters

Katharina E.J. Jungnickel [1], Joanne L. Parker[1] & Simon Newstead [1]

Amino acids play essential roles in cell biology as regulators of metabolic pathways. Arginine in particular is a major signalling molecule inside the cell, being a precursor for both l-ornithine and nitric oxide (NO) synthesis and a key regulator of the mTORC1 pathway. In mammals, cellular arginine availability is determined by members of the solute carrier (SLC) 7 family of cationic amino acid transporters. Whereas CAT-1 functions to supply cationic amino acids for cellular metabolism, CAT-2A and -2B are required for macrophage activation and play important roles in regulating inflammation. Here, we present the crystal structure of a close homologue of the mammalian CAT transporters that reveals how these proteins specifically recognise arginine. Our structural and functional data provide a model for cationic amino acid transport in mammalian cells and reveals mechanistic insights into proton-coupled, sodium-independent amino acid transport in the wider APC superfamily.

[1] Department of Biochemistry, University of Oxford, South Parks Road, Oxford OX1 3QU, UK. Correspondence and requests for materials should be addressed to S.N. (email: simon.newstead@bioch.ox.ac.uk)

Arginine plays an important role in regulating cellular metabolism through its ability to activate the mTORC1 kinase[1,2], a central signalling hub that integrates diverse environmental inputs to coordinate anabolic and catabolic processes in cells, such as insulin release[3] and intestinal stem cell migration[4]. In immune cells, arginine can also be shunted towards the production of nitric oxide (NO) via the action of nitric oxide synthase 2, where it functions in the production of reactive oxygen species[5]. The intracellular concentration of arginine is therefore an important factor in determining key aspects of mammalian physiology[6].

Cellular arginine availability is determined in part by the efficiency and capacity of arginine transporters expressed in the plasma membrane, and in mammals is mostly mediated by members of the solute carrier 7 family (SLC7, pfam00324). The SLC7 family forms part of the much larger amino acid polyamine cation (APC) superfamily, which is found in all organisms and represents a large family of secondary active transport proteins responsible for uniport, symport and antiport of a broad range of cellular metabolites into and out of the cells[7,8].

Several inherited disorders are linked to mutations in members of the SLC7 family, including cystinuria[9] and lysinuric protein intolerance[10], both of which lead to abnormal amino acid transport, growth defects and kidney disease. Several prescription drugs, including L-DOPA and gabapentin are also recognised by members of the SLC7 family, which are expressed in the blood brain barrier and influence the transport and retention of drug molecules into the central nervous system[11–13].

The SLC7 family is divided into two subgroups, the cationic amino acid transporters (CATs, SLC7A1−4 and SLC7A14)[14] and the L-type amino acid transporters (LATs, SLC7A5-13 and SLC7A15)[15]. Unlike the CATs, which function as monomers in the plasma membrane, the LATs are obligate heterodimers, forming a disulphide linked dimer with a single transmembrane spanning glycoprotein, designated SLC3, which functions to traffic the transporter to the plasma membrane and aid in protein stability[15]. Despite arising from the same ancestral transporter, the LAT and CAT families show various differences, not just in the interaction with the SLC3 family, but also with respect to their substrate specificity and transport mechanism[16]. HATs transport a wide range of zwitterionic amino acids and for high affinity transport couple uptake to the $Na^+$ electrochemical gradient[17]. CATs in contrast are specific for cationic amino acids[14]. Originally designated as system y[+], the CATs mediate $Na^+$- independent uptake with high affinity ($K_M$ in the micromolar range) and are believed to operate as either exchangers or facilitators[14].

To regulate the uptake of arginine in different tissue and cell types the CAT-2 transporter is also found in two isoforms, CAT-2A and -2B, which differ in their affinity for arginine. Whereas CAT-2B, which is expressed in macrophages and T-cells, exhibits high affinity (μM) for arginine, CAT-2A, which is expressed in liver and muscle cells, displays low affinity (mM)[14]. Although the difference between CAT-2A and -2B has been localised to two amino acids that reside on an intracellular loop[18], there exists no molecular description for how this seemingly minor change can affect transport function in these proteins.

Although in mammals CATs operate as exchanger or facilitators, in plants they are pH dependent[19], raising the question of whether in certain species these systems are proton coupled. Indeed, the SLC7 family forms part of the much larger APC superfamily of secondary active transporters and are distantly related to the SLC36 family of proton coupled amino acid transporters[7,20]. A key question in the field of transport biology is how superfamilies, such as the APC, have evolved to switch between passive facilitator, antiporter and coupled symporter mechanisms.

Crystal structures of APC superfamily transporters have revealed a conserved '5 + 5 inverted topology' fold, wherein the first five TM helices are related to the second five helices via a pseudo two-fold symmetry axis running parallel to the plane of the membrane[21]. Although originally discovered in $Na^+$ coupled symporters[22], the same fold was also found associated with a sodium independent amino acid symporter ApcT[23] and the amino acid exchangers AdiC[24] and GadC[25]. AdiC, an arginine-agmatine exchanger, has been used as a model system for understanding amino acid recognition in the large-neutral amino acid transporter, LAT-1 (SLC7A5)[26] and -2 (SLC7A8)[27,28]. However, neither AdiC nor the other currently reported structures from APC family members share high sequence identity with the mammalian CATs, leaving fundamental questions concerning amino acid recognition, selectivity and transport mechanism unanswered.

To address these questions, we determined the crystal structure of a close prokaryotic SLC7 homologue in complex with bound L-alanine and L-arginine ligands and studied its transport mechanism using in vitro biochemical assays. We exploited the close homology to the human proteins to identify a single amino acid substitution that dictates arginine recognition in the CATs. Site directed mutagenesis combined with liposome-based transport assays further revealed a structural explanation for the kinetic differences between CAT-2A and CAT-2B, and suggest a model for cationic amino acid recognition in eukaryotic cells.

## Results

**Structure of GkApcT.** Following an extensive screen of prokaryotic homologues with high (>40%) amino acid sequence identity to eukaryotic SLC7 transport proteins, we identified a suitable candidate from the thermophilic bacterium *Geobacillus Kaustophilus*, GkApcT (Supplementary Fig. 1A & B). Initial attempts to crystallise GkApcT in the lipidic cubic phase were unsuccessful. Crystals were eventually obtained in the presence of L-Alanine and cholesterol doped monoolein, and diffracted X-ray to 2.86 Å. Phases were determined using molecular replacement, employing a CHAINSAW model of the distantly related APC homologue MjApcT from *Methanococcus Jannaschii* (PDB 3GIA)[23]. The side chain register was further assigned and validated using the anomalous signal from a sulphur-SAD data set collected at 2.7 Å wavelength (Supplementary Fig. 2A & B) and the final structure refined to an $R_{factor}/R_{free}$ of 21.5/27.0 % (Table 1).

GkApcT consists of 12 transmembrane helices (TMs), which adopt the canonical APC superfamily fold, wherein transmembrane helices TM1−TM5 are related to helices TM6−TM10 by a pseudo two-fold symmetry axis located in the plane of the membrane[29] (Fig. 1). Both TM1 and TM6 are broken in the centre of the transporter to form two discontinuous helices, termed here 1A, 1B and 6A and 6B respectively (Supplementary Fig. 1C). In GkApcT the two additional helices, TM11 and TM12, wrap around and to one side of the transporter. The N-terminus folds into a lateral helix that packs tightly on the cytoplasmic side of the protein before folding into the first TM helix. Our sequence alignment suggests a similar helix may exist in the mammalian proteins (Supplementary Fig. 1).

The electron density maps also revealed the presence of an additional single transmembrane helix that packs against TM5, and orientated with its amino-terminus facing the cytoplasm. The maps were of sufficient quality to allow us to identify this helix as YneM, which has recently been renamed as MgtS (Supplementary Fig. 2C & D), a single spanning transmembrane protein from *Escherichia coli*. MgtS packs tightly against GkApcT creating a hydrophobic pocket between TM5 of the transporter. In our structure, we observed density for a cholesterol molecule in this

**Table 1 Data collection and refinement statistics for GkApcT**

| | GkApcT WT + L-Ala PDB: 5OQT | GkApcT WT + L-Ala | GkApcT-M321S + L-Arg PDB: 6F34 |
|---|---|---|---|
| *Data collection* | | | |
| Space group | $P2_12_12_1$ | $P2_12_12_1$ | $P2_12_12_1$ |
| Cell dimensions | | | |
| $a, b, c$ (Å) | 76.9, 83.15, 119.74 | 77.19, 83.41, 119.64 | 77.04, 82.70, 118.78 |
| $\alpha, \beta, \gamma$ (°) | 90.0, 90.0, 90.0 | 90.0, 90.0, 90.0 | 90.0, 90.0, 90.0 |
| Wavelength (Å) | 0.9686 | 2.7 | 0.980 |
| Resolution (Å)[a] | 68.3-2.86 (2.93-2.86) | 48.28-3.30 (3.56) | 64.63-3.13 (3.21-3.13) |
| $R_{merge}$ (%) | 15.2 (178.9) | 34.2 (377.9) | 17.3 (145.2) |
| $R_{pim}$ (%) | 7.7 (84.2) | 7.9 (85.6) | 7.4 (85.3) |
| $I/\sigma I$ | 6.3 (1.1) | 11.1 (1.1) | 5.5 (1.1) |
| CC1/2 (%) | 86.3 (51.8) | 99.6 (45.8) | 99.8 (67.9) |
| Completeness (%) | 99 (100) | 98.9 (97.7) | 99.9 (99.8) |
| Multiplicity | 5.2 (5.4) | 20.1 (21.4) | 6.4 (6.3) |
| Anom. completeness (%) | | 99.0 (97.8) | |
| Anom. multiplicity | | 10.9 (11.3) | |
| Mid-slope | | 1.112 | |
| *Refinement* | | | |
| Resolution (Å) | 64.72-2.86 (2.96-2.86) | | 50.93-3.13 (3.24-3.13) |
| No. unique reflections | 18190 | | 13867 |
| $R_{work}/R_{free}$ (%) | 21.5/27.1 | | 23.1/26.2 |
| Ramachandran favoured | 95.0 | | 92.0 |
| Ramachandran outliers | 0.63 | | 0.84 |
| R.m.s deviations | | | |
| Bond lengths (Å) | 0.01 | | 0.002 |
| Bond angles (°) | 1.10 | | 0.473 |

[a]Highest resolution shell is shown in parentheses

pocket (Supplementary Fig. 2E). Interestingly cholesterol was essential for the crystallisation of GkApcT and the observed interaction in the crystal structure would suggest a role in stabilising the interaction with MgtS. A similar role in stabilisation of MgtA, a P-type ATPase $Mg^{2+}$ transporter, was recently reported for MgtS[30]. We hypothesised that the combination of MgtS and cholesterol in our structure may point to a role in stabilisation of GkApcT. Using circular dichroism to measure the thermal stability of GkApcT we determined the melting temperature of the GkApcT-MgtS heterodimer to be 57 °C (Supplementary Fig. 3A). In the presence of cholesterol hemisuccinate (CHS) the melting temperature was increased to 72 °C, showing a clear thermal stabilising effect. Lipids did not appear to stabilise the transporter, as shown in a bicelle control. To test the effect of the MgtS interaction we constructed a ΔMgtS knock out strain of *E. coli* C43 (DE3) and used this for expression. The melting temperature of the GkApcT in the absence of MgtS was 58 °C, similar to that observed for the GkApcT-MgtS heterodimer (Supplementary Fig. 3B). However, unlike for the GkApcT-MgtS heterodimer we observed no change in the presence of CHS.

We also observed that the uptake of $^3$H-L-Alanine (L-Ala) in the protein purified from the ΔMgtS knock out strain transported less substrate than the C43 (DE3) cells, despite the initial rates being similar (Supplementary Fig. 3C & D). It is possible that the stability provided through the MgtS interaction enables GkApcT to operate over a longer period of time. Why we observe cholesterol in our crystal structure is also an interesting question. Although bacteria do not contain cholesterol, certain species do contain similar cyclic molecules termed hopanoids[31], which function similarly to cholesterol in regulating the fluidity of the membrane in these species[32]. It is possible that the cholesterol we observe in GkApcT may be a surrogate for a hopanoid binding site in vivo and suggests that bacterial transport function may be regulated by hopanoid binding in a similar way to their eukaryotic counterparts[33].

**Amino acid binding site**. GkApcT adopts a ligand bound inward occluded state with solvent filled cavities extending towards the bound ligand from either side of the membrane (Fig. 2a). Whereas the extracellular side of the transporter is tightly sealed through interactions between TM1, TM6 against TM3, TM8, the intracellular side is noticeably more open, with a long water filled tunnel leading from the cytoplasm to the amino acid binding site (Fig. 2b). Clear electron density was observed in the structure for the bound ligand, L-Ala (Fig. 2c). The amino acid is accommodated within a small cavity located towards the extracellular side of the membrane. A conserved aromatic side chain on TM6, Phe231 and threonine, Thr43, on TM1 provide the main steric occlusion that seals the binding site from the extracellular side of the membrane. In contrast, intracellular access is less obstructed, with only a thin occlusion separating the ligand binding site from the solvent filled cytoplasmic tunnel leading to the inside of the cell. This thin gate is constructed principally from hydrophobic side chains on TM8, Met321, and TM6A, Ile234 and Glu115 on TM3 (Fig. 2b). The packing of TM6B against TM3 is mediated through well-coordinated water molecules, which are bound between Glu115 and Asp237 on TM6B (Fig. 2d).

Within the binding site the L-Ala ligand makes a number of polar interactions to backbone atoms of the protein. The amino group predominantly interacts with the carbonyl groups of Phe231, Ala232 and Ile234 from TM6 with one interaction to Ile40 from TM1. In contrast, the carboxyl group interacts with TM1, making polar contacts to the amino groups of Thr43 and Gly44. Interestingly a further interaction is made to either a water molecule or sodium ion, which is bound in a small pocket in the unwound region of TM1. Several members of the HAT family are proposed to co-transport neutral amino acids with sodium ions[17] and the APC fold is also used in sodium coupled transporters, such as LeuT[22]. The water molecule or sodium ion appears to stabilise the unwound region of TM1 through several hydrogen bonds to the backbone atoms of the protein, with a further

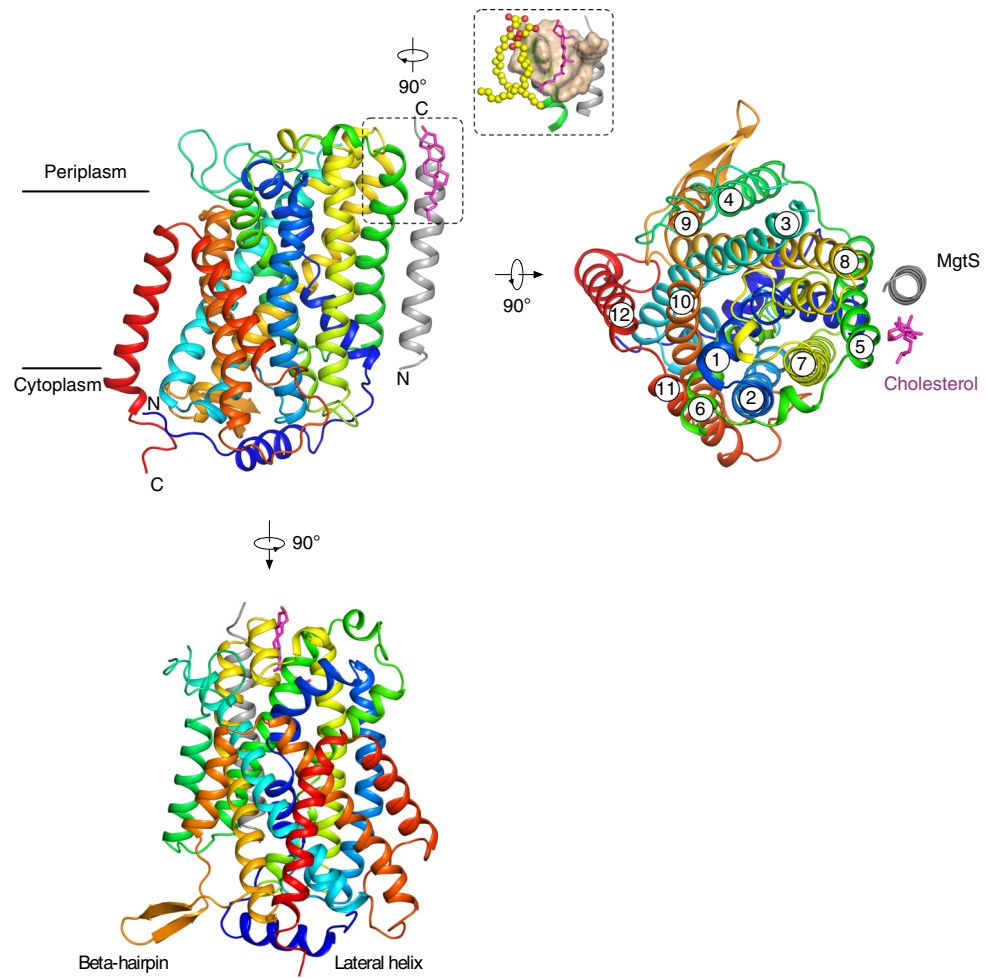

**Fig. 1** Crystal structure of GkApcT. Crystal structure of GkApcT-MgtS heterodimer at 2.86 Å. Helices are coloured blue to red from the N-termini. MgtS is coloured grey and the bound cholesterol pink. Inset, a close-up view of the cholesterol binding site made between GkApcT and MgtS. Two monoolein lipids are also observed close to the cholesterol binding site and shown as spheres

interaction made to a conserved tyrosine on TM7, Tyr268 (Fig. 2e). Interestingly, the hydroxyl group of Tyr268 sits in the same position as the sodium ion in the Na 1 site in LeuT[34] (Supplementary Fig. 4). This raised the question of whether GkApcT was coupled to either the sodium or proton electrochemical gradients.

**GkApcT is a proton coupled amino acid transporter**. Although mammalian CATs are facilitators, plant members of the SLC7 family are pH-dependent[19] and members of the SLC36 family of amino acid transporters are proton-coupled[20]. To deepen our understanding of amino acid transport and identify molecular commonalities between GkApcT and the wider APC superfamily, we sought to further understand the transport mechanism by studying [$^3$H]-L-Ala uptake under the influence of inwardly directed proton or sodium gradients. GkApcT was determined to be proton-coupled, with uptake of L-[$^3$H]-Ala observed in the presence of a membrane potential, negative inside and a pH gradient, alkaline inside (Fig. 3a). Transport was sodium-independent and abolished in the presence of the proton ionophore CCCP, indicating GkApcT is a proton-coupled amino acid transporter.

A previous study on the mechanism of a sodium-independent pH activated amino acid facilitator, MjApcT, suggested a semi-conserved lysine on TM5 as a site of proton binding[23]. GkApcT

contains an equivalent lysine on TM5, Lys191 (Fig. 3b). Similar to MjApcT, the amine group of Lys191 occupies the same position as the Na2 site in LeuT coordinating the discontinuous region of TM1 with TM8 (Supplementary Fig. 4). The pKa of Lys191 is 7.3 in the current structure[35], suggesting the amine group is in equilibrium between the neutral and positively charged form. To test whether Lys191 was responsible for proton coupling in GkApcT, we mutated this residue to alanine, arginine and asparagine (Fig. 3c). Under electrogenic uptake, which allows substrate accumulation, and counterflow uptake, which allows substrate equilibration[36], activity was abolished with the alanine and arginine substitutions. However, proton-driven transport was detectable, albeit at a lower level, in the Lys191Asn variant. This result suggests that Lys191 is not required for proton-coupled transport in GkApcT. However, when no transport is observed for a variant this may indicate disruption to substrate recognition. To test whether this was the case here, we used a thermal stability assay to investigate the effect of these variants on the ability of GkApcT to recognise L-Ala. We observed that in the presence of 10 mM L-Ala the WT protein showed a noticeable increase in melting temperature from ~58 to ~69 °C (Supplementary Fig. 5A). A similar stabilising effect was also observed in the Lys191Ala, Arg and Asn variants (Supplementary Fig. 5B), indicating that the loss of transport function we observed is not due to an inability to bind ligand but rather indicates that Lys191 is required for another aspect of the transport mechanism.

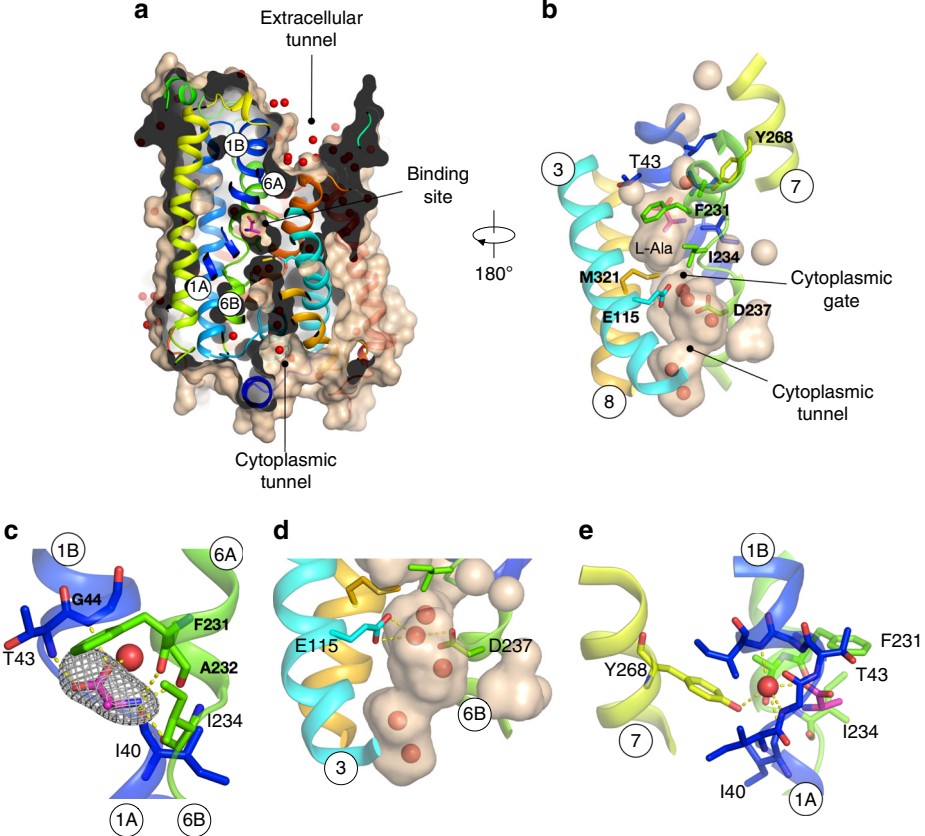

**Fig. 2** Structure of the amino acid binding site. **a** Molecular surface representation of GkApcT showing the bound L-Ala ligand and position of the solvent filled extracellular and cytoplasmic tunnels with water shown as red spheres. **b** Zoomed in view of the ligand binding site, showing the location of waters in the cytoplasmic tunnel (brown shading) and location of the cytoplasmic thin gate. **c** View of the bound L-Ala ligand, showing the mFo-DFc difference electron density map contoured at 3σ, grey. **d** Zoomed in view of the cytoplasmic tunnel, showing the interaction between the acidic residues in TM3 and 6. **e** View of the water molecule coordinating the α-carboxylate group of the L-Ala ligand

It is highly likely that TM1 undergoes significant conformational change during transport, as discussed below, and the neutral form of the lysine side chain would facilitate a dynamic interaction with the carbonyl groups in the unwound region of TM1. We therefore suggest that Lys191 has an important role in stabilising the unwound region of TM1 during transport, acting as a surrogate for the sodium ion observed in the Na2 site in structural homologues such as LeuT. The asparagine substitution in GkApcT at this position is able to partially substitute for this interaction. Taken together our data suggests that Lys191 may undergo protonation and deprotonation during the transport cycle; however, this is not involved in coupling transport to the proton electrochemical gradient but rather is important to facilitate the conformational changes in this region of the transporter.

To identify the proton-coupling sites, we next focused on two acidic residues, Glu115 (TM3) and Asp237 (TM6) that sit directly below the bound L-Ala ligand (Fig. 2b, d). As discussed above, both Glu115 and Asp237 coordinate an interaction between TM3 and TM6 via a water molecule. Asp237 is highly conserved within the SLC7 family (Supplementary Fig. 1) including those members that are not proton coupled. This suggests Asp237 is not the site of proton binding in GkApcT. However, its mutation to alanine did result in complete loss of function (Fig. 3d), highlighting an important role in transport. Mutation of Glu115 to alanine also resulted in loss of transport activity; however, a glutamine at this position demonstrated counterflow activity only, indicating a role in proton coupling. Such a role is consistent with its pKa value of

8.22 in the crystal structure, whereas Asp237 on the other hand has a pKa value of 5.48. Interestingly, the Glu115Gln variant can still recognise alanine to the same extent as wild-type protein (Supplementary Fig. 5C), suggesting the reduction of activity observed in the counterflow assay is not the result of the loss of the ability to bind ligand, but indicates an important secondary role in the transport cycle. It is likely given the observed interaction between Glu115 and Asp237 that this side chain coordinates interactions between TM3 and TM6A, which as discussed below forms part of the intracellular gate of the transporter.

**Amino acid selectivity mechanism.** CATs are highly specialised for L-arginine, L-lysine and L-ornithine transport unlike their close homologues the LAT transporters, which can also recognise neutral and anionic amino acids[15]. Given the important role that CATs play in NO signalling and macrophage activation[37,38], we wished to understand the molecular basis of arginine selectivity within the CAT family. During our attempts to crystallise GkApcT, we used a counterflow assay to establish the protein as a relatively broad specificity amino acid transporter with a preference for small hydrophobic and polar amino acids (Fig. 4a). It was surprising, given the high sequence identity to the CATs, that GkApcT showed poor recognition of the cationic amino acids L-arginine and L-lysine. Analysis of the binding site in GkApcT revealed that several side chains that sit close to the bound L-Ala are considerably smaller in the human CAT homologues

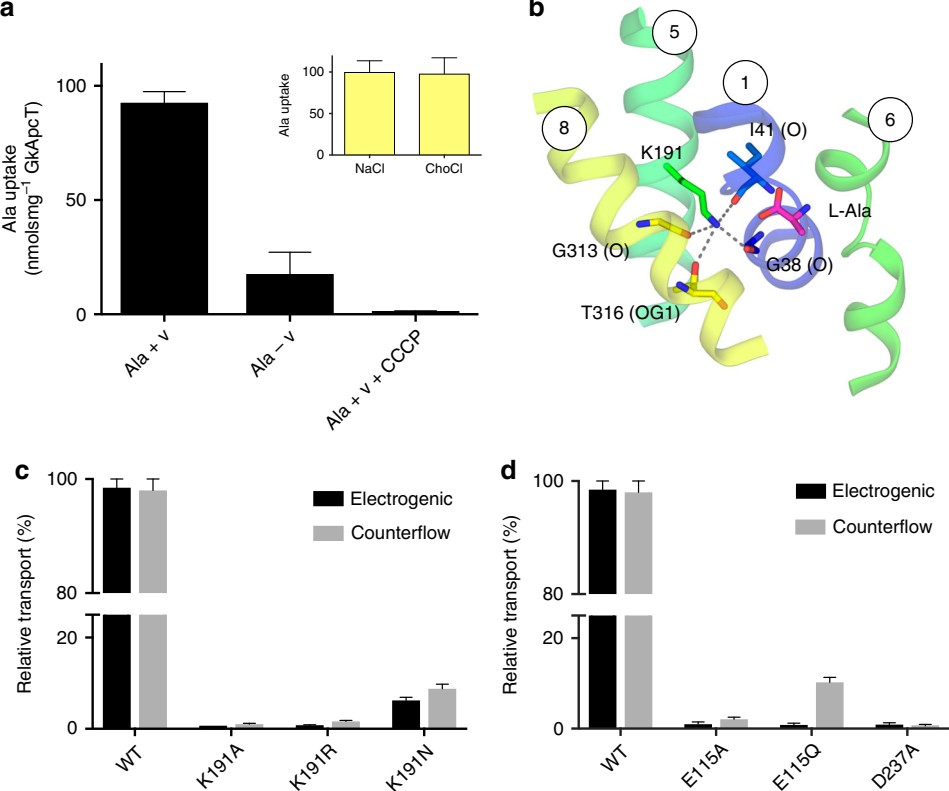

**Fig. 3** Functional characterisation of proton coupling residues in GkApcT. **a** Proton-coupled uptake assays showing that accumulation of L-Ala is driven by an inwardly directed proton electrochemical gradient generated using valinomycin (+v) and an inwardly directed pH gradient (−v). Transport was abolished in the presence of the proton ionophore CCCP. Inset—transport is independent of sodium. **b** View of Lys191 in the crystal structure, highlighting the interactions made to TMs 1 and 8. The bound L-Ala ligand is shown in magenta. **c** Mutational analysis of Lys191 using electrogenic and counterflow accumulation of [3]H L-Ala. **d** Mutational analysis of Glu115 and Asp237 using electrogenic and counterflow accumulation assays. $n = 3$ independent experiments, error bars s.d.

(Fig. 4b). In particular, Met321, Ala119, Val317 and Ile234 all adopt smaller side chains in the human proteins, being serine, glycine, serine and valine respectively. Given that amino acid recognition in GkApcT is mediated via backbone interactions to the α-carboxy and amino groups, we speculated that the substrate specificity, was due to steric constraints within the binding site, which effectively acted to exclude larger amino acids. To assess whether these side chains had any impact on specificity we generated a series of point mutants to assess their relative contribution to substrate specificity (Fig. 4c). Ile234 is positioned in the unwound region of TM6 and forms part of the cytoplasmic gate (Fig. 2b). Mutation to valine severely reduced transport in this variant. Ala119 and Val317 on TM3 and 8 respectively are positioned close to the bound L-Ala ligand, forming one side of the binding pocket. In a similar manner to Ile234, these mutants also showed reduced transport. However, we discovered that Met321Ser was able to recognise and transport both L-Arg and L-Lys to a greater extent than the wild-type protein (Fig. 4c). Indeed, the affinity of this variant for arginine is significantly increased as demonstrated by a reduction in the $IC_{50}$ value from 6 mM in the WT to 11.4 µM in the M321S variant (Fig. 4d). To further validate this result, we obtained a crystal structure at 3.13 Å of the Met321Ser variant with bound L-Arg (Fig. 4e and Table 1). The structure shows that replacement of Met321 with serine does not affect the position of the helices within the protein, with a r.m.s.d. of 1.15 Å over 480 $C_\alpha$ atoms when compared to the WT structure. As expected, replacement of Met321 with serine substantially opens up the ligand binding site facilitating the increased size of the L-Arg ligand. The L-Arg is positioned

with the α-carboxy and amino termini making identical interactions to those observed with L-Ala, with the side chain extending down into the space opened up by removal of the methionine. In our structure, the guanidinium group sits close to both Glu115 and Asp237. In the mammalian CATs Asp237 is strictly conserved, suggesting that this interaction may be important for cationic amino acid selectivity, whereas Glu115 is replaced with a serine, which would still provide a favourable polar interaction.

**Isoform differences between CAT-2A and -2B.** An intriguing aspect of CAT biochemistry is the observed kinetic difference between the two isoforms of CAT2. While CAT-2B displays the key features that characterise the system y+ transporters, of µM affinity and trans stimulation[39,40], CAT-2A has ten-fold lower substrate affinity and shows no trans stimulation effect[41]. The mechanistic basis underpinning these differences in transport kinetics however is currently not well understood, owing in part to the lack of an appropriate structural model. Previous studies have identified the intracellular region between TM8 and 9 as being important for the transport differences observed between CAT-2A and -2B and identified two amino acids responsible for the low affinity of CAT-2A, Arg369 and Asn381, which was inserted into this region extending the sequence by one amino acid[18]. Arg369 in CAT-2A is equivalent to Arg334 in GkApcT, which in CAT-2B, CAT-1 and CAT-3 is a glutamate–aspartate pair (Fig. 5a). To further understand the molecular basis for the affinity switch we mutated the equivalent residue in GkApcT,

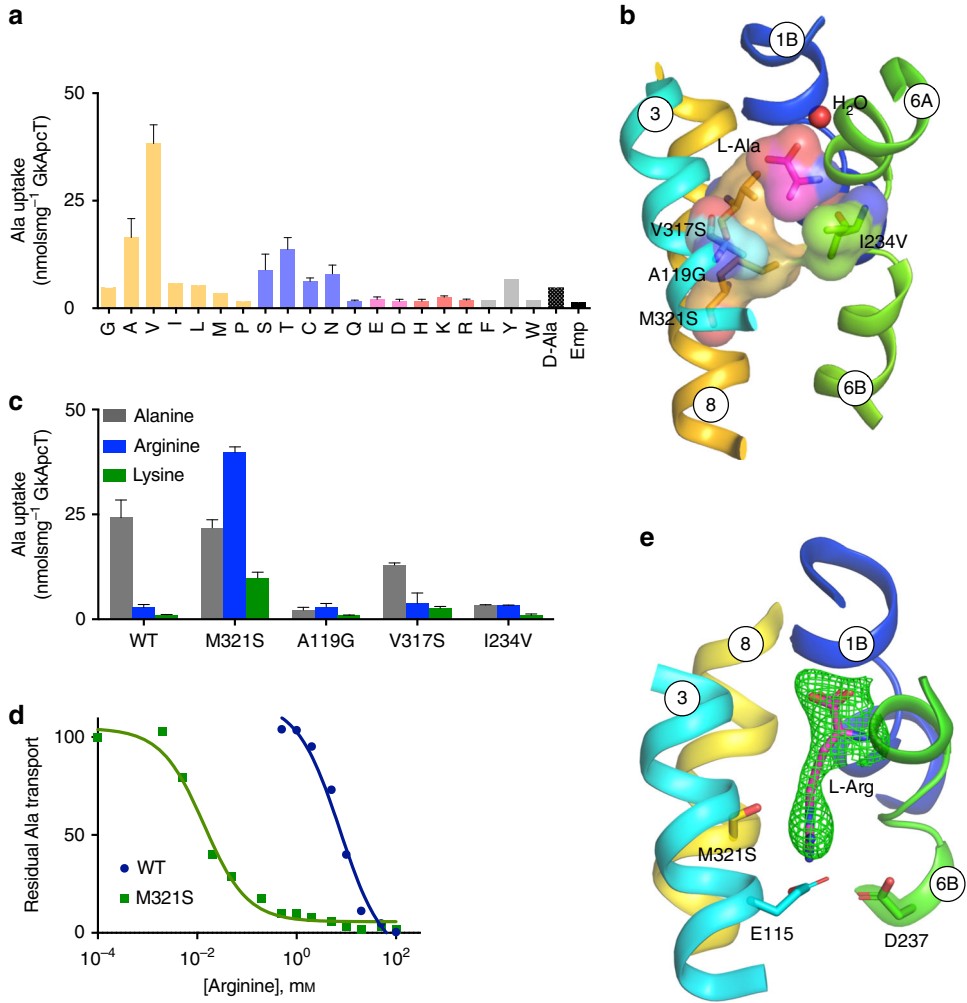

**Fig. 4** Functional characterisation of the amino acid binding site. **a** Substrate specificity of GkApcT analysed using counterflow of the 20 canonical amino acids plus D-Ala and a water control (emp) using external $^3$H L-Ala. $n = 3$ independent experiments, error bars s.d. **b** View showing the molecular volume of the side chains within the binding site of the WT GkApcT structure. Due to the size of Met321, there is no space to accommodate the extra side chain bulk of L-Arg. **c** Effect of mutating residues in the binding site to their counterparts in CAT-2A. Accumulation was measured using $^3$H L-Ala uptake using counterflow accumulation with internal L-Ala, L-Arg or L-lys. $n = 3$ independent experiments, error bars s.d. **d** Crystal structure of the Met321Ser variant with bound Arginine (purple), shown in sticks. The mFo-DFc difference electron density map contoured at 3 σ, (green mesh). **e** Representative IC$_{50}$ curves determined for L-Arg binding to either the WT or M321S variant of GkApcT using an electrogenic uptake assay. IC$_{50}$ values were calculated from three independent replicates

Arg334 to glutamate. Remarkably we observed that the IC$_{50}$ value for L-Ala uptake reduced from 2 μM in the WT protein to 0.4 μM in the Arg334Glu variant, mimicking the affinity increase observed for the CAT-2B isoform (Fig. 5b). The crystal structure provides a rational explanation for this affinity change. As noted above, the binding site is closed to the cytoplasmic side of the membrane through the packing of TM6B with TM3 and TM8. Arg334 makes a salt bridge interaction to the lateral helix through Glu15, which would stabilise TM8 and through side chain interactions TM6B and TM3. We suggest that changing this residue to glutamate, as occurs in the high-affinity CAT-2B isoform, causes a structural change in TM8, which in turn alters the packing of TM6B and TM3 affecting the affinity of amino acids in the binding site.

## Discussion

The mammalian CAT proteins share only ~20% amino acid sequence identity with their counterparts, the L-type amino acid transporters. Unlike the LATs, the CATs function as amino acid

facilitators and are sodium-independent[14], raising the question of how this subfamily of SLC7 transporters preferentially recognise and transport cationic amino acids into the cell. Previous studies on amino acid selectivity in AdiC, a bacterial homologue of the L-type proteins[26], suggests that arginine selectivity is driven by general electrostatic interactions in the binding site, rather than through specific interactions to side chain residues[42]. Indeed, an overlay of the arginine-bound AdiC structure reveals several common features in the binding site between AdiC and GkApcT (Supplementary Fig. 6A). In particular, the positively charged α-amino group interacts similarly with carbonyl groups in the discontinuous region of TM1 and TM6. The aromatic side chain of Phe231 also sits in a similar position to a conserved tryptophan in AdiC, Trp202, which acts to close the binding site from the extracellular side of the membrane during transport[43]. A notable difference however is observed for the negatively charged α-carboxylate group. In AdiC the α-carboxylate group accepts two H-bonds from the side chain Ser26 (TM1) and the amide nitrogen of Gly27. In contrast, in GkApcT we observe the α-carboxylate group of L-Ala interacting with a well-ordered water

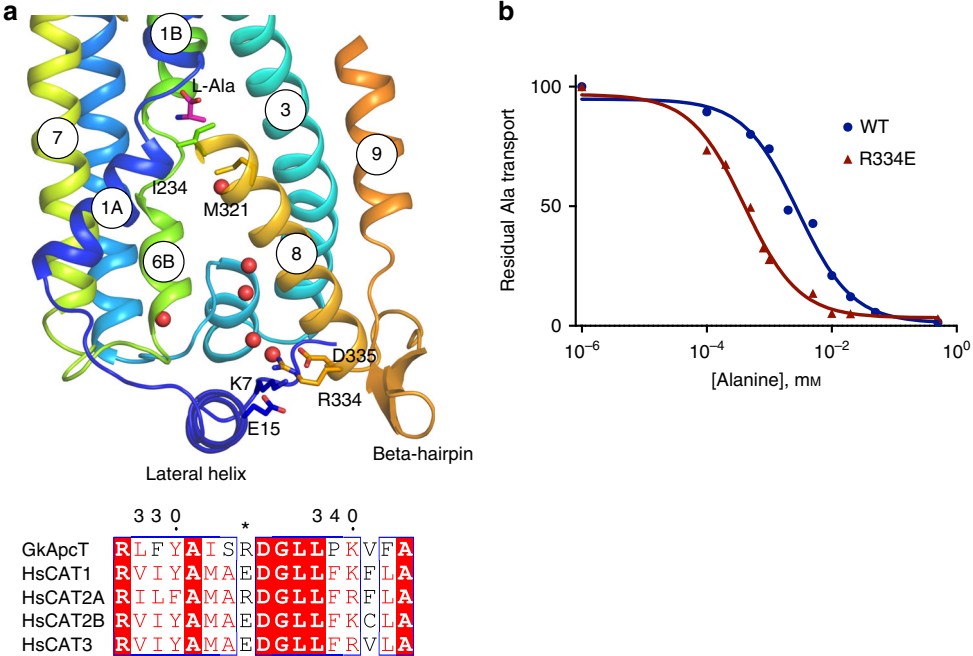

**Fig. 5** Structural basis for isoform difference between CAT-2A and -2B. **a** Zoomed-in view of the cytoplasmic region of the transporter, highlighting the interactions between TM6B, TM8 and the lateral helix. A sequence alignment of TM8 highlighting the difference (*) between CAT-2A and CAT-1, -2B and -3 is shown. **b** Representative IC$_{50}$ curves determined for L-Ala binding to either the WT or R334E variant of GkApcT using electrogenic accumulation assays. IC$_{50}$ values were calculated from three independent replicates

molecule, in addition to the backbone amide groups of Thr43 and Gly44 (Fig. 2b, d). However, the most striking feature of this arrangement is the water molecule sits in a similar position to the Na 1 ion in LeuT, coordinating the interaction between the substrate and TM1 (Supplementary Fig. 4)[34]. We modelled this density as water, due to the sodium independence of GkApcT in our transport assays. The use of water as a coordinating molecule is a feature often associated with transiently formed complexes, and suggests movement of TM1 would cause disruption to the hydrogen bond network and weakening of the interactions to the amino acid. Dynamic water networks have been proposed to play an important role in coordinating TM3 and TM10 in AdiC[44]. However, the role of water in coordinating the L-Ala ligand in GkApcT suggests a more direct mechanistic link between water coordination and ligand binding. It is interesting to note that System y$^+$L transporters, exemplified by y$^+$LAT-1 (SLC7A7), utilise sodium for the transport of neutral amino acids but dispense with the ion for cationic substrates[14,17]. It appears from our structure that this site might be exchangeable between sodium and water in these systems, as sodium can also bind to proteins with five-coordinate geometry[45].

In AdiC the guanidinium group of the bound arginine and its decarboxylated form, agmatine, are observed interacting with a conserved aromatic side chain on TM8, Trp293 through a cation-pi interaction and to several side chains on TM3[43,44] (Supplementary Fig. 6B). Interestingly, Trp293 in AdiC overlays onto Met321 in GkApcT, which our data also show has an important role in determining substrate selectivity (Fig. 4c). However, the overlay of AdiC and GkApcT suggests that the guanidinium group of arginine will be accommodated in a different position than observed in AdiC, due to TM3 being positioned further into the binding pocket, pushing the side chain towards the conserved serine in the mammalian proteins (Supplementary Fig. 6C).

Secondary active transporters operate via an alternating access mechanism, whereby the protein switches between outward facing and inward facing states to move ligands across the

membrane (Fig. 6a)[46]. Within the APC superfamily several hypotheses have emerged regarding the structural rearrangements[47]. Certainly, individual families have their own idiosyncratic mechanisms for transport, but within the 'LeuT-like' fold it is generally considered that TM1 and 6 act as toggle switches, which orchestrate the conformational changes during the alternating access cycle[21]. In our case, GkApcT is the first proton-coupled amino acid transporter to be structurally and biochemically studied, revealing important insights into the transport mechanism in this subfamily of SLC7 transporters. In the current inward occluded state the bound L-Ala ligand is trapped inside the binding site through the formation of a thick extracellular gate and thin intracellular gate (Fig. 6b). For the bound ligand to exit into the cell, either TM1A or TM6B must swing away from TM8, breaking the cytoplasmic thin gate (Figs. 2a, 6a). The tethering of TM1A to the lateral helix would likely severely reduce the movement of this helix, supporting a more dynamic role for TM6B. In the crystal structure, we see that TM6B makes several interactions that help close the cytoplasmic gate. In particular a water-mediated interaction between Asp237 (TM6B) and Glu155 (TM3), and Thr241 (TM6B) and Arg327 (TM8). Our data suggest that Glu115 plays important roles in proton binding in GkApcT. Given the pKa of this residue, ~8, in the current structure, we suggest that Glu115 is protonated, and interacting with Asp237 via the observed water molecule (Fig. 2c). Proton release from Glu115 would facilitate the movement of TM6B away from TM3 and TM8, driven by the repulsion between the two acidic residues. This movement would serve to break the interactions between the carbonyl groups of TM6 with the α-amino group of the ligand, thus dissolving the binding site, a process no doubt aided through the use of water as a coordinating molecule to the ligand and TM1. The importance of the interaction between TM6 and 8 is further supported by our finding that the isoform differences between CAT-2A and -2B are likely due, in part, to the stability of TM8, which may have an effect on the kinetics of transport by modulating this movement.

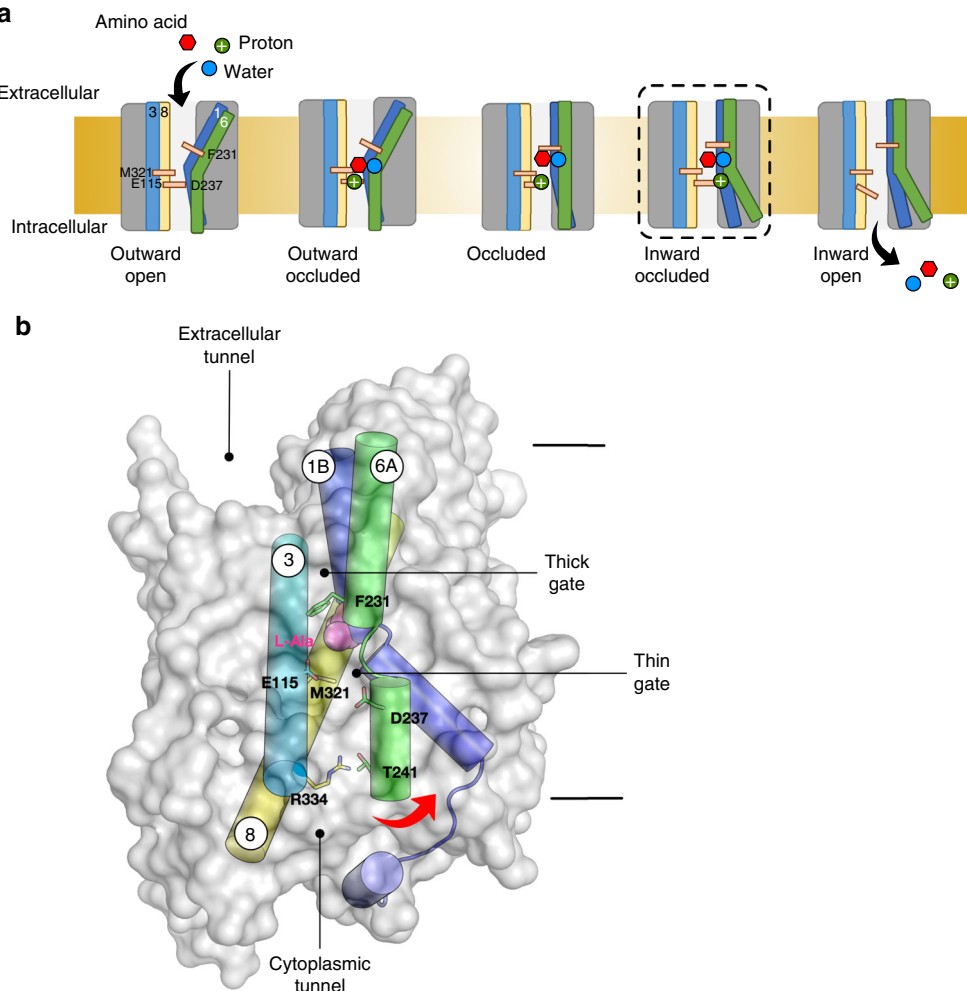

**Fig. 6** A model for alternating access transport in GkApcT. **a** Schematic representation of alternating access transport in GkApcT. Starting from an outward facing state the transporter must reside in a conformation that facilitates access of protons, water and amino acid to the binding site. Binding of amino acid causes conformational changes in TM1 and 6 that act to close the extracellular gate. This may be facilitated by water binding to TM1 and movement of F231 that facilitates formation of the occluded state. Further conformational changes are predicted to occur that move the transporter into a fully occluded state, before the intracellular gate starts to open[46]. Our crystal structure likely represents this 'inward occluded' state, where the intracellular gate is still present, but held shut through side chain interactions rather than main chain packing. In GkApcT E115 is likely protonated in this state, holding TM6 in close approximation to TM8, where M321 forms part of the intracellular gate. Proton release from E115 causes the water mediated interaction to D237 to break, and TM6 to swing away from TM3 and 8, opening the binding site to the inside of the cell and releasing the bound amino acid and possibly a water molecule. **b** Structure of GkApcT shown in molecular surface representation with the predicted gating helices shown. Our data suggest that movement of TM6 away from TM3 and 8 opens the binding site to the interior of the cell, and facilitates release of the bound amino acid

A long-standing question in the SLC7 field is the position of the heavy subunit of HATs (rBAT and 4F2hc; SLC3A1-A2) relative to the LAT domain (SLC7A5-13)[15]. The current structure of MgtS in complex with GkApcT reveals how a single TM helix can pack against the transport domain of an APC family member. The heteromeric complex between GkApcT and MgtS is consistent with a previous homology model of the SLC7A6 complex, which places the single TM of the heavy subunit (4F2hc) in close proximity to TM5 in LAT-2[27] and may therefore represent a viable structural model for this interaction. A recent report on the modulation of LAT-1 transporter activity by cholesterol suggests a role for the sterol molecule in stabilising the interaction with SLC3A2[48]. Our observation that the interaction between MgtS and GkApcT creates a cholesterol binding site (Fig. 1) and that this also serves to stabilise the heterodimer, suggests a similar structural arrangement may also occur between the heavy and light subunits of the HATs.

In summary, we present crystal structures of the first close homologue of the mammalian CATs. The complexes with bound alanine and arginine reveal the molecular basis for amino acid selectivity is based on steric restrains rather than charge complementarity, providing insight into how the SLC7 family has evolved to select specific sub-sets of amino acids. An important question in transport biology is how similar protein folds have evolved different transport mechanisms. The structural and functional data reported here reveal how the '5 + 5 inverted topology' fold, which is widespread in many SLC families, has adapted to couple amino acid uptake to the proton electrochemical gradient within the APC superfamily.

## Methods

**Protein expression and purification**. The gene encoding GkApcT (Uniprot Q5L1G5) was amplified from *Geobacillus kaustophilus* genomic DNA and cloned into the pWaldo-GFP vector using primers detailed in Supplementary Table 1[49].

Variant forms of GkApcT were made through site directed mutagenesis using standard methods. Wild-type and variant GkApcT proteins were expressed and purified to homogeneity using standard Ni[2+]-affinity protocols[50]. Membranes were thawed and solubilised in purification buffer which consisted of, 1× PBS containing an additional 150 mM NaCl and 1% n-dodecyl⁻β-D-maltopyranoside (DDM) with stirring for 1 h. The solubilised material was recovered through ultracentrifugation at >200,000×g for 1 h. A final concentration of 15 mM imidazole was added and the protein was bound to nickel resin (GE Healthcare) in batch for between 1 and 2 h, or when the level of binding had plateaued, as measured using the GFP counts. The resin was washed with purification buffer containing 25 mM imidazole and 0.1% DDM for 20–30 column volumes. GkApcT was eluted from the resin with 250 mM imidazole. TEV protease was added and the protein was dialysed overnight in gel filtration buffer containing 0.03% DDM (20 mM Tris pH 7.5, 150 mM NaCl). After dialysis, the protein was passed through a HisTrap column to remove the TEV protease and the GFP tag. The pure protein was concentrated and applied to a superdex 200 10/300 gel filtration column (GE Healthcare). For crystallisation, size exclusion chromatography was performed using a buffer consisting of 20 mM Tris-HCl and 150 mM NaCl with 0.03% DDM detergent (crystallisation grade, Glycon), and the protein was concentrated to 20 mg ml$^{-1}$ and stored at −80 °C. For functional characterisation, the detergent was exchanged through size exclusion chromatography using a Superdex 200 10/300 column into 20 mM Tris-HCl, 150 mM NaCl and 0.3% n-decyl-β-D-maltopyranoside (DM).

**Crystallisation.** Crystallisation was performed using protein at 10 mg ml$^{-1}$ final concentration, as determined using absorbance at 280 nm. L-Alanine or L-arginine were added to a final concentration of 10 mM and the protein was left on ice for at least 2 h prior to LCP set up. Protein-laden mesophase was obtained by mixing cholesterol doped monoolein (10% CS)[51] with protein in a 60:40 (w/w) ratio using a coupled syringe device (Art Robbins, USA). Initial crystals appeared at 20 °C in a condition of the MemMeso screen (Molecular Dimensions Ltd, UK), which was further optimised to 28–34% PEG 400, 0.1 M sodium acetate pH 4.0 and 0.1 M potassium fluoride, containing 10 mM of either alanine or arginine. Diamond or rod-shaped crystals appeared after 3 days and reached their full size (~30 μm in the longest dimension) after 2 weeks. Only the rod-shaped crystals diffracted to high resolution, with the diamond-shaped crystals being noticeably less well ordered. Wells were opened using a tungsten glasscutter and the crystals were harvested using 30 μm micromounts (MiTeGen). Crystals were cryo-cooled directly in liquid nitrogen and stored in unipucks.

**Structure determination.** X-ray diffraction data collected at beamline I24, Diamond Light Source, UK resulted in a complete data set of GkApcT to 2.86 Å. Indexing and integration were performed with XIA2 using the DIALS pipeline[52–54], followed by scaling and merging with AIMLESS[55]. Initial phases were obtained by molecular replacement (MR) using PHASER in the CCP4 suite[56]. The template used for MR was generated using CHAINSAW with the homologous MjApcT structure (PDB: 3GIA). Due to the potential for phase bias in the phasing procedure, anomalous data were collected at both I23, Diamond Light Source and BL1-A, Photon factory, Japan at 2.7 Å using inverse beam data collection. Due to crystal size the smaller beam size at BL-1A gave higher resolution data. The data were processed in XDS[57] and combined using XDS_SCALE, treating the Friedel-pairs individually. The data were prepared with SHELXC[58] and the anomalous difference Fourier map calculated with ANODE[59]. Model building into the electron density map was performed in COOT[60], with structure refinement carried out in BUSTER2.10.2. Geometry restraints for ligands were calculated using the grade server supplied by Global Phasing Ltd. Model validation was carried out using the Molprobity server[61]. Images were prepared using PyMol[62].

**Point mutations and *E. coli* C43 MgtS knock-out strain.** Site-directed mutagenesis was performed using standard PCR methods and employing the primers detailed in Supplementary Table 1. Protein variants were purified as described for the WT protein. C43(DE3) ΔMgtS strain was generated using standard procedures and employing the gene doctoring method[63,64]. In short, 300 bp sections up- and downstream of the *yneM* gene were cloned into the pDOC-K vector into cloning region 1 and 2 (CR1 and CR2), respectively, flanking the kanamycin cassette. The pDOC-K vector was transformed via electroporation into the C43(DE3) *E. coli* strain, used for expression of *Gk*ApcT. Activation of the pABSCE vector, transformed into the same cells, induced recombination of the kanamycin cassette into the gene locus of *yneM*. Cells were plated onto LB agar plates containing kanamycin. To check for the loss of the pABSCE vector a colony was spread onto LB agar plates with chloramphenicol expecting no colonies to grow in presence of this antibiotic. The KO strain was then cured from the kanamycin resistance according to ref. [64]. Successful gene knock-out was verified by PCR.

**Protein reconstitution into liposomes.** Purified wild-type and variant proteins were reconstituted into lipid vesicles using the dilution method[65]. In brief, lipid vesicles consisting of a 3:1 ratio of POPE:POPG (Avanti Polar Lipids, USA) in lipid buffer (50 mM potassium phosphate, pH 7.0) were mixed with protein in DM gel filtration buffer at a 60:1 lipid:protein ratio (wt:wt). The lipid protein mix was incubated for 1 h before being diluted rapidly into a large volume of lipid buffer.

Proteoliposomes were harvested by ultracentrifugation and dialysed overnight in lipid buffer. The proteoliposomes were stored at −80 °C.

**Proton-coupled and counterflow transport assays.** For proton-driven uptake, proteoliposomes were thawed, centrifuged at 90,000×g for 30 min at 4 °C and re-suspended in internal buffer (20 mM potassium phosphate pH 6.50, 100 mM potassium acetate, 2 mM magnesium sulphate) followed by five freeze-thaw cycles and extrusion through a 400 nm polycarbonate filter. Proteoliposomes were subsequently diluted 1:25 (v/v) into external buffer containing 250 nM ³H-Ala with 100 μM unlabelled alanine (20 mM sodium phosphate pH 6.50, 100 mM sodium chloride, 2 mM magnesium sulphate) with 10 μM valinomycin. Assays were performed at 25 °C and stopped through rapid filtration onto 0.22 μm nitrocellulose filters. The filters were washed under vacuum with 2 × 2 ml 0.1 M lithium chloride prior to scintillation counting. The ³H signal was converted to molar concentrations of peptide using standard curves for each substrate. For ligand-driven counterflow transport[66], proteoliposomes were loaded through repeated freeze thaw cycles with 25 mM citrate phosphate, pH 6.0, 100 mM sodium chloride, 2 mM magnesium sulphate and 2 mM of ligand for determining the substrate specificity (Fig. 4a) or 10 mM L-Ala or Arg or Lys for subsequent counterflow studies (Fig. 4c). Transport was initiated following a 1:25 (v/v) dilution into 25 mM citrate phosphate, pH 6.0, 100 mM sodium chloride, 2 mM magnesium sulphate containing 100 μM L-Ala and 250 nM ³H L-Ala.

**Transport assays to determine IC$_{50}$ values.** Transport was carried out at 25 °C by diluting GkApcT proteoliposomes 1:25 into external buffer (20 mM sodium phosphate pH 6.50, 100 mM sodium chloride, 2 mM magnesium sulphate with 10 μM valinomycin) containing 250 nM ³H-Ala, and increasing concentrations of competitor. Reactions were terminated after 1 min by transferring 50-μl aliquots of each reaction mixture into 2 ml cold lithium chloride. Reaction mixtures were filtered and analysed as previously described. The entire experiment was conducted in triplicate to obtain a mean and standard deviation.

**Thermal stability measurements.** To evaluate thermal stability of protein samples two methods were employed. The first used CD spectroscopy to calculate melting temperatures[67]. Concentrated (>10 mg ml$^{-1}$) protein was diluted to ~0.1 mg ml$^{-1}$ into buffer consisting of 10 mM potassium phosphate pH 6.5, 50 mM sodium sulphate and 3× CMC of the desired detergent. CD spectra were obtained between 20 and 85 °C at 2 °C increments on a JASCO 815 spectrophotometer. Stability curves were obtained by plotting the absorption value at $\lambda = 220$ nm (% helicity) against the temperature. The resultant data were plotted in Prism v7 (GraphPad Software) and analysed using a sigmoidal curve fit. In addition, thermal stability in the presence and absence of 10 mM ligand was also analysed using a Prometheus NT.48 (NanoTemper Technologies). The proteins were diluted to a final concentration of 0.5 mg ml$^{-1}$ into buffer containing 25 mM citrate phosphate, pH 6.0, 100 mM sodium chloride, 2 mM magnesium sulphate and 0.03% (w/v) DDM. Thermal measurements were carried out in a range from 20 to 95 °C with 1 °C per min steps. The resulting melting curves were generated by plotting the first derivative of the fluorescence ratio at 330 nm/350 nm against temperature.

**Data availability.** Atomic coordinates for the crystal structures have been deposited in the Protein Data Bank under accession numbers 5OQT (WT, L-Ala co-crystal complex) and 6F34 (M321S, L-Arginine co-crystal complex). Other data are available from the corresponding author upon reasonable request.

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

## Acknowledgements

We thank the staff of beamlines I24 and I23 Diamond Light Source, UK, Proxima-2A, Soleil, France and BL-1A, Photon Factory, Japan for assistance. We thank Professor O. Nureki and Professor R. Ishitani, University of Tokyo, Japan, for assistance in accessing BL-1A under the approval of the Photon Factory Program Advisory Committee (Proposal 2016-R01) and Dr. Armin Wagner, Diamond Light Source, UK, for assistance in accessing I23 during commissioning. K.E.J.J. wishes to thank Professor Elspeth Garman and Dr. Ed Lowe for training and assistance in crystallographic work. This work was supported by Wellcome (102890/Z/13/Z), the European commission Marie Curie Training Network NanoMem and Molecular Dimensions Ltd.

## Author contributions

S.N. conceived the study. K.E.J.J., J.L.P. and S.N. designed the experiments and interpreted the data. J.L.P. and S.N. wrote the paper with assistance from K.E.J.J.

## Additional information

**Competing interests:** The authors declare no competing financial interests.

