## [Peer Review File · Nature Communications]

Reviewers' comments:

Reviewer #1 (Remarks to the Author):

The manuscript of Jungnickel et al. describes the structural and functional characterizations of GkApcT, a prokaryotic homologue of mammalian cationic amino acid transporters (CATs; part of solute carrier family 7). The SLC7 family can be divided into the CATs and the LATs (L-type amino acid transporters). While several structures have been determined for LAT homologues, a structure for a CAT homologue has remained elusive thus far. The GkApcT structure reveals the anticipated global architecture of the Amino acid Polyamine Cation (APC) superfamily consisting of an inverted repeat of five TMs, but in addition highlights structural features specific to this family such as a detailed description of the substrate binding site. Remarkably, GkApcT was co-crystallized with a single spanning transmembrane protein from *E. coli*, YneM and the presence of YneM is suggested to impact protein stability by stabilizing a cholesterol molecule. GkApcT is demonstrated to be a proton-coupled Alanine transporter. Initial indications for residues involved in proton-coupling are presented. Mutagenesis of residues involved in the substrate specificity demonstrates that binding of the typical CAT substrates arginine and lysine in GkApcT is prevented by sterical constraints mostly involving Met321.

The major findings of this manuscript are the first structure of a close homolog of the cationic amino acid transporters, its functional characterization and initial studies on its coupling mechanism and substrate specificity.

The paper is well written and its findings are novel and of interest to the community and the wider field.

Major remarks.

P5, transport (..) was lower; this is an unclear statement. Does lower reflect to the transport rate or to the accumulation level? Based on the data in ed fig 3c the initial rate seems affected only a bit, while the accumulation level seems about half. The latter does not necessarily represent a feature of the protein, but might as well reflect a different diameter (and thereby: internal volume) of the proteoliposomes. The latter could be checked by adding a protonophore to dissipate the proton gradient. In addition, a reconstitution control is missing for indicating that similar amounts of protein were functionally reconstituted (e.g., in the form of a coomassie-stained SDS-PAGE sample).

Fig3c, the decreased transport (rate/accumulation??) of the mutants can have several causes beyond their potential involvement in proton coupling, e.g. in affecting substrate binding indirectly. To exclude this, control experiments should involve the counterflow transport assay used for fig 3d or a substrate binding assay.

Fig3d, given the suggested major role of glu115 in proton coupling, the authors should make a more extensive effort to explain the decreased counterflow transport rate. Is substrate binding affected by E115Q as well? Or is the turnover rate affected?

P21, the final statement is formulated too strong, how proton coupling is achieved is not (yet) revealed due to the complex response of the glu115 mutants. Coupled transport is not exclusively affected, also counterflow transport is suggesting a more diverse mechanistic role of this residue.

YneM has been recently renamed to MgtS (see Wang et al, 2017, PMID: 28512220). The observations on GkApcT should be discussed in the light of the observations made for MgtS.

Minor remarks.

P2, for high affinity transport couple.. there is no correlation between the coupling ion and the affinity for the substrate in secondary transporters, the authors probably meant to relate the use of a coupling ion to the accumulation of the substrate instead. Please correct.

Fig1, the content of the inset is not clear. What does the wheat-colored surface density represent?

YneM has been recently renamed to MgtS (see Wang et al, 2017, PMID: 28512220).

Fig2b, labels partly hidden

ED fig 3 has no legend

Fig3A, it is unclear whether accumulation levels or initial rates are depicted.

Fig3C/d, Fig 4ACE, specify transport mode and whether initial rates or accumulation levels were used
Reconstitution controls are missing for indicating that similar amounts of protein were functionally reconstituted (e.g., in the form of a coomassie-stained SDS-PAGE sample).

P24, the protein reconstitution section is duplicated

Fig 4A, what do the stars indicate?

Reviewer #2 (Remarks to the Author):

The manuscript by Jungnickel, Parker and Newstead reports crystal structures and functional studies of a bacterial amino acid transporter that shares high homology to the human solute carrier SLC 7 family of cationic amino acid transporters. The crystals were grown in lipidic cubic phase and solved to a good resolution for a membrane protein. The authors demonstrated that the GkApcT can transport amino acids with preferences to alanine, valine and a few other neutral amino acids. The transport is driven by a proton gradient. The authors then used the structure to examine mechanistic questions with regard to proton translocation and substrate selectivity. Two residues, K191 and E115, are candidate for mediating the proton translocation based on their location and on studies of equivalent residues on other homologs. Mutational studies led the authors to conclude that K191 does not mediate proton binding while E115 could. For substrate selectivity, the authors focused on ways to increase transport of arginine. They found that M321 to S increases the rate of arginine transport significantly. The crystal structure of M321S was then solved and the binding site has an electron density that could be interpreted as an arginine. Finally, the authors identified a residue Arg 334 in GkApcT that is equivalent to Arg 369 in the human CAT2A and glutamate in human Cat 2b. When Arg 334 was mutated to a glutamate, alanine uptake is altered and the direction of change seems to mirror the difference between CAT2A and CAT2B.

The crystal structures of GkApcT are of very good quality, and it is also very interesting that YneM was co-purified and resolved in the structure. In addition, the mechanistic questions that the authors tried to address are all of high significance. I have several concerns that are mostly on data interpretation.

First, assignment of alanine and arginine to the electron densities at the putative binding site does not seem unambiguous. Are the densities for the substrate amino acids comparable to these of surrounding residues?

Second, converting electrogenic uptake to substrate affinity may not be accurate. A proper binding assay is required.

Third, in a counter flow assay shown in Figure 4A, what does it mean when valine produces the largest alanine flow? In the end, a counter flow assay is a quick way to get a rough estimate on substrate selectivity and it is suitable for initial characterizations. It would be more proper if an uptake assay is used for at least the amino acids that are relevant in this study, arginine, lysine, alanine and valine to name a few.

Forth, a proper binding assay should be performed to compare substrate affinity of M321S and wt.

Fifth, the rate of arginine transport by M321S is more than 100 fold slower than that of alanine by the wild type. Does M321S binds or transports other amino acids better? And if other amino acids are

transported faster, can the author still claim that M321S converts the substrate selectivity of the binding pocket?

Sixth, both K191N and E115Q mutations have such low activity compared to that of wild type, it is not justified to claim that K is not responsible for proton binding while E is. One could simply interpret the data as both residues are involved in proton coupling.

While I fully appreciate the difficulty of obtaining the structures, tailoring functional assays to the transporter, and using a bacterial transporter as a step stone to reach for understanding of human transporters, I feel that the questions the authors chose to address would require more rigorous experiments and interpretations.

We would like to thank both referees for their critical reading of the manuscript and helpful comments. We were pleased to see that the referee's appreciated the significant insights this work provides into understanding the structural basis for amino acid transport in the SLC7 family. In light of these comments, we have revised the manuscript, which now includes a higher resolution crystal structure for the M321S L-Arg bound complex (3.1 Å) and several additional biochemical experiments, which further support our conclusions and clarify the points in the paper. We address specific concerns below with the reviewer's comments in blue and our response in black.

Reviewer #1:

Major remarks.

1. P5, transport (..) was lower; this is an unclear statement. Does lower reflect to the transport rate or to the accumulation level? Based on the data in ed fig 3c the initial rate seems affected only a bit, while the accumulation level seems about half. The latter does not necessarily represent a feature of the protein, but might as well reflect a different diameter (and thereby: internal volume) of the proteoliposomes. The latter could be checked by adding a protonophore to dissipate the proton gradient. In addition, a reconstitution control is missing for indicating that similar amounts of protein were functionally reconstituted (e.g., in the form of a coomassie-stained SDS-PAGE sample).

We agree with the reviewer that this statement was ambiguous. We have now performed further experiments that demonstrate the initial rates are the same for both GkApcT with and without MgtS (YneM was recently renamed MgtS and we have now changed this in our Ms to be consistent with the literature). We have included this data as Extended Data Figure 3D and added further explanation of why we think the accumulation in the protein purified in the absence of MgtS accumulated L-Ala to a lower level.

"We also observed that the uptake of ³H-L-Alanine (L-Ala) in the protein purified from the ΔMgtS knock out strain transported less substrate than the C43 (DE3) cells, despite the initial rates being similar (ED Figure 3C & D). It is possible that the stability provided through the MgtS interaction enables GkApcT to operate over a longer period of time."

We have also included reconstitution controls for all mutants used in the study (Extended Data Figure 5D). Referee 1 also suggested whether the volume of the liposomes may be different. In these experiments, our liposomes were prepared in the same manner, requiring extrusion through a 400 nM filter. They will therefore not differ substantially between the two samples.

2. Fig3c, the decreased transport (rate/accumulation??) of the mutants can have several causes beyond their potential involvement in proton coupling, e.g. in affecting substrate binding indirectly. To exclude this, control experiments should involve the counterflow transport assay used for fig 3d or a substrate binding assay.

In light of the reviewers comments we have now expanded our analysis of the lysine₁₉₁ mutants used in Figure 3C. We have included counterflow data of the mutants that shows that only the Lys₁₉₁Asn variant retains any transport activity. We have also used a thermal stability assay to assess the ability of the ligand (L-Ala) to stabilise these variants. Our new data, which we present in ED Figure 5, clearly shows that these variants can still recognise substrate as ligand binding results in a significant stabilisation effect. In light of these new data, we have significantly altered the text of the manuscript to better describe our findings.

"A previous study on the mechanism of a sodium-independent pH activated amino acid facilitator, MjApcT, suggested a semi-conserved lysine on TM₅ as a site of proton binding {Shaffer, 2009}. GkApcT contains an equivalent lysine on TM₅, Lys₁₉₁ (Figure 3B). Similar to MjApcT, the amine group of Lys₁₉₁ occupies the same position as the Na₂ site in LeuT coordinating the discontinuous region of TM₁ with TM₈ (ED Figure 4). The pK_a of Lys₁₉₁ is 7.3 in the current structure {Olsson, 2011}, suggesting the amine group is in equilibrium between the neutral and positively charged form. To test whether Lys₁₉₁ was responsible for proton coupling in GkApcT, we mutated this residue to alanine, arginine and asparagine (Figure 3C). Under electrogenic uptake, which indicates coupled transport, and counterflow uptake, which indicates uncoupled transport {Kaback, 2001}, activity was abolished with the alanine and arginine substitutions. However, proton driven transport was detectable, albeit at a lower level, in the Lys₁₉₁Asn variant. This result suggests that Lys₁₉₁ is not required for proton coupled transport in GkApcT. However, when no transport is observed for a variant this may indicate disruption to substrate recognition. To test whether this was the case here, we used a thermal stability assay to investigate the effect of these variants on the ability of GkApcT to recognise L-Ala. We observed that in the presence of 10 mM L-Ala the WT protein showed a noticeable increase in melting temperature from ~ 58 to ~ 69 °C (ED Figure 5A). A similar stabilising effect was also observed in the Lys₁₉₁Ala, Arg and Asn variants (ED Figure 5B), indicating that the loss of transport function we observed is not due to an inability to bind ligand but rather indicates that Lys₁₉₁ is required for another aspect of the transport mechanism.

It is highly likely that TM₁ undergoes significant conformational change during transport, as discussed below, and the neutral form of the lysine side chain would facilitate a dynamic interaction with the carbonyl groups in the unwound region of TM₁. We therefore suggest that Lys₁₉₁ has an important role in stabilising the unwound region of TM₁ during transport, acting as a surrogate for the sodium ion observed in the Na₂ site in structural homologues such as LeuT. The asparagine substitution in GkApcT at this position is able to partially substitute for this interaction. Taken together our data suggests that Lys₁₉₁ may undergo protonation and deprotonation during the transport cycle, however this is not involved in coupling transport to the proton electrochemical gradient but rather is important to facilitate the conformational changes in this region of the transporter."

3. Fig3d, given the suggested major role of glu₁₁₅ in proton coupling, the authors should make a more extensive effort to explain the decreased counterflow transport rate. Is substrate binding affected by E₁₁₅Q as well? Or is the turnover rate affected?

We agree with the referee that the role of Glu₁₁₅ is very interesting, and we make clear in our paper that our current hypothesis is that this residue plays an important role in coupling

protonation to the conformation of the intracellular gate. Our evidence for this conclusion comes from the crystal structure, where we observe Glu115 making a water mediated interaction to Asp237 on TM6B. Specifically, the referee wished us to investigate the reduced transport activity we observed for Glu115Qln. Using a thermal stability assay we show that E115Q is stabilised to the same extent as WT protein (ED Figure 5), which shows that the reduced transport we observe in the counterflow assay is not due to an inability to recognise ligand. We therefore suggest that the reduced activity is the result of a disruption of the interaction that we observe the crystal structure to Asp237. Clearly Glu115 is playing a role in both protonation and conformational change. We have now included this data and description in the revised Ms.

"Mutation of Glu115 to alanine also resulted in loss of transport activity, however a glutamine at this position demonstrated counterflow activity only, indicating a role in proton coupling. Such a role is consistent with its pKa value of 8.22 in the crystal structure, whereas Asp237 on the other hand has a pKa value of 5.48. Interestingly, the Glu115Gln variant can still recognise alanine to the same extent as wild type protein (ED Figure 5C), suggesting the reduction of activity observed in the counterflow assay is not the result of the loss of the ability to bind ligand, but indicates an important secondary role in the transport cycle. It is likely given the observed interaction between Glu115 and Asp237, that this side chain coordinates interactions between TM3 and TM6A, which as discussed below forms part of the intracellular gate of the transporter. "

4.P21, the final statement is formulated to strong, how proton coupling is achieved is not (yet) revealed due to the complex response of the glu115 mutants. Coupled transport is not exclusively affected, also counterflow transport is suggesting a more diverse mechanistic role of this residue.

We agree with the referee that the role of Glu115 is complex and as discussed above likely to link protonation to conformational changes in the transporter. It is highly likely that additional are also involved in protonation and conformational change however, and future studies will seek to identify these. However, we feel that the current data do demonstrate that Glu115 is the most likely candidate for proton binding given the current crystal structure and the results from the assays.

5.YneM has been recently renamed to MgtS (see Wang et al, 2017, PMID: 28512220). The observations on GkApcT should be discussed in the light of the observations made for MgtS.

We have now changed this in the Ms.

Minor remarks.

P2, for high affinity transport couple.. there is no correlation between the coupling ion and

the affinity for the substrate in secondary transporters, the authors probably meant to relate the use of a coupling ion to the accumulation of the substrate instead. Please correct.

We are simply using the main conclusions from the original study by Dr E, Closs 'Closs EI, Boissel JP, Habermeier A, Rotmann A. Structure and function of cationic amino acid transporters (CATs). *J Membr Biol* **213**, 67-77 (2006)', which used the same terminology. We suspect this is due to a change in the kinetics of transport when using sodium as the coupling ion. The 2006 paper suggests, to us, that members of the LAT family can also use the sodium electrochemical gradient in addition to acting as facilitators.

Fig1, the content of the inset is not clear. What does the wheat-coloured surface density represent?

Corrected and now referred to.

YneM has been recently renamed to MgtS (see Wang et al, 2017, PMID: 28512220). Done

Fig2b, labels partly hidden – done.

ED fig 3 has no legend – corrected.

Fig3A, it is unclear whether accumulation levels or initial rates are depicted. – accumulation levels were used and have been specified in the materials and methods.

Fig3C/d, Fig 4ACE, specify transport mode and whether initial rates or accumulation levels were used. – accumulation levels were used and have been specified in the materials and methods.

Reconstitution controls are missing for indicating that similar amounts of protein were functionally reconstituted (e.g., in the form of a coomassie-stained SDS-PAGE sample). – we have now included an SDS-PAGE gel showing the reconstituted protein. (ED Figure X).

Fig 4A, what do the stars indicate? – we have removed the stars.

Reviewer #2 (Remarks to the Author):

1. Assignment of alanine and arginine to the electron densities at the putative binding site does not seem unambiguous. Are the densities for the substrate amino acids comparable to these of surrounding residues?

The density for the L-Ala ligand in the 2.8 Å structure is similar to those of the surrounding ligands, with equivalent B factors. We agree with the referee that the density for the L-Arginine amino acid in the original M321S complex structure was not as good as we would

have liked. During the revision, we screened additional crystals at Proxima-2a, which has an Eiger 9M detector, and managed to collect a new data set at 3.1 Å (PDB 6F34 – please see new deposition report). We have now included this in our revised Ms and hope the referee will agree that the new difference density is now unambiguous for L-Arginine.

2. Converting electrogenic uptake to substrate affinity may not be accurate. A proper binding assay is required.

Whilst we certainly agree that an IC_{50} is not a K_D , it nevertheless does show how the transporter responds to changing concentration levels and therefore can be used to infer affinity. We therefore feel that this assay is a valuable metric for understanding how a transporter recognises different ligands. For the purposes of the current study we feel that the IC_{50} experiments are sufficient to support our conclusions concerning the ability of GkApcT to recognise different amino acids.

3. In a counter flow assay shown in Figure 4A, what does it mean when valine produces the largest alanine flow? In the end, a counter flow assay is a quick way to get a rough estimate on substrate selectivity and it is suitable for initial characterizations. It would be more proper if an uptake assay is used for at least the amino acids that are relevant in this study, arginine, lysine, alanine and valine to name a few.

We agree that counterflow assays are a qualitative way to show the substrate specificity of a transporter. We feel this is the ideal assay to assess the substrate specificity of GkApcT and the variants around the binding site. However, we do not agree that for this study it is necessary to undertake additional experiments to assess the kinetics of transport for the different amino acids. We feel that additional kinetic studies on these variants is more suitable for a follow up study and importantly, would not add much to the present discussion.

4. A proper binding assay should be performed to compare substrate affinity of M321S and wt.

The counterflow data shown in Figure 4C demonstrates that the M321S variant is sufficient for the recognition and transport of L-Arginine. Indeed, our ability to trap a complex of M321S with L-Arginine in the crystal structure testifies to the ability of this mutation to confer L-Arginine specificity. Our IC_{50} curves clearly show that the M321S variant is able to recognise L-Arginine at much lower concentrations, which is likely the result of higher affinity.

5. The rate of arginine transport by M321S is more than 100 fold slower than that of alanine by the wild type. Does M321S bind or transport other amino acids better? And if other amino acids are transported faster, can the author still claim that M321S converts the substrate selectivity of the binding pocket?

We agree with the referee that the electrogenic uptake of L-Alanine by M321S was significantly impaired relative to WT. Considering the location of Met321 in the binding site sitting above Glu115 and Asp237, it is not surprising that removal of this side chain has a negative effect on electrogenic transport. However, we feel that this data distracted from the main findings in our paper (and most importantly did not add any additional insight) and have therefore decided to remove this.

6. Both K191N and E115Q mutations have such low activity compared to that of wild type, it is not justified to claim that K is not responsible for proton binding while E is. One could simply interpret the data as both residues are involved in proton coupling.

We have adjusted the text to more fully explain our hypotheses regarding the role of K191 and E115. Our data provide strong enough evidence that K191 is unlikely to couple transport to the proton electrochemical gradient. If this was the case we should not see any transport in the K191N, and we do, albeit at a low level. We certainly agree that the role of E115 is complex, and would expect that as we study this system further we identify additional sites of proton binding. However, we feel that our conclusions are justified in so far as highlighting that K191 is unlikely to be involved in coupling transport to proton movement, and that E115 is.

While I fully appreciate the difficulty of obtaining the structures, tailoring functional assays to the transporter, and using a bacterial transporter as a step stone to reach for understanding of human transporters, I feel that the questions the authors chose to address would require more rigorous experiments and interpretations.

We feel this statement is unnecessarily critical and not a fair reflection on the insights gained through this work. This manuscript reports the **first crystal structures** for a bacterial homologue of the mammalian CAT transporters and also the **first proton coupled amino acid transporter**. We have shown in both our crystal structures and through the biochemical assays that arginine specificity is achieved through a change of a **single amino acid** – which is remarkable. Reducing the IC_{50} for the protein from 6mM to ~110μM is undeniably a significant difference, especially given the counterflow data that shows that the wild type protein cannot transport L-Arginine! In addition, the data presented in Figure 5 also provides important **mechanistic insights into the isoform differences** between CAT-2A and -2B. Given these insights we feel that the present study more than adequately supports our conclusions regarding the transport mechanism in the SLC7 family.

REVIEWERS' COMMENTS:

Reviewer #1 (Remarks to the Author):

My comments have been addressed appropriately and I suggest acceptance for publication. I have few minor comments mostly concerning some phrasings.

Concerning my original comment 1. P5, transport (..) was lower; this is an unclear statement. Does lower reflect to the transport rate or to the accumulation level?

>This point has been appropriately addressed. The explanation of this effect remains difficult and will be subject for scientific discourse beyond the scope of this manuscript. The suggestion that MgtS enables GkApcT to operate over a longer period of time I find unlikely: the total procedure starting with the purification and ending with the membrane reconstitution takes at least 24 hrs, and thus it seems unlikely that the protein without MgtS would be inactivated in the last 9 minutes (and in the friendly environment of the lipid membrane). I still favor the hypothesis that the internal volume is slightly different: a 25% difference in radius would already result in a two-fold difference in internal volume. Despite the state-of-the-art extrusion approach of the authors, such differences could easily occur in my opinion. This small disagreement with the authors does not decrease the importance and high overall quality of the paper though.

Concerning my original comment 2. Fig3c, the decreased transport (rate/accumulation??) of the mutants can have several causes beyond their potential involvement in proton coupling...

>The sentence "Under electrogenic uptake, which indicates coupled transport, and counterflow uptake, which indicates uncoupled transport {Kaback, 2001}, "

Please reformulate, the current phrasing suggests that protons are not involved in the counterflow transport mode, while they are. They just do not drive the direction of transport. The sentence should read: "electrogenic uptake, which allows substrate accumulation, ..counterflow uptake, which allows substrate equilibration"

Concerning my minor remark "P2, for high affinity transport couple.. there is no correlation between the coupling ion and .." and the authors comment:

>The reference indicated at the end of this statement (17) refers to a different paper (Pfeiffer et al. 1999), please correct. The effect is interesting. It may also reflect cooperativity in the Na- and amino acid binding sites.

Concerning minor remark "Fig3A, it is unclear whether accumulation levels or initial rates are depicted. – accumulation levels were used and have been specified in the materials and methods. "

>This should be specified in the legend as well.

Concerning minor remark "Fig3C/d, Fig 4ACE, specify transport mode and whether initial rates or accumulation levels were used. – accumulation levels were used and have been specified in the materials and methods. "

>This should be specified in the legend as well.

Response to referee's comments.

My comments have been addressed appropriately and I suggest acceptance for publication. I have few minor comments mostly concerning some phrasings.

Concerning my original comment 1. P5, transport (..) was lower; this is an unclear statement. Does lower reflect to the transport rate or to the accumulation level?

*>This point has been appropriately addressed. The explanation of this effect remains difficult and will be subject for scientific discourse beyond the scope of this manuscript. The suggestion that MgtS enables GkApcT to operate over a longer period of time I find unlikely: the total procedure starting with the purification and ending with the membrane reconstitution takes at least 24 hrs, and thus it seems unlikely that the protein without MgtS would be inactivated in the last 9 minutes (and in the friendly environment of the lipid membrane). I still favor the hypothesis that the internal volume is slightly different: a 25% difference in radius would already result in a two-fold difference in internal volume. Despite the state-of-the-art extrusion approach of the authors, such differences could easily occur in my opinion. **This small disagreement with the authors does not decrease the importance and high overall quality of the paper though.***

No need to address, no doubt we can debate at the next conference!

Concerning my original comment 2. Fig3c, the decreased transport (rate/accumulation??) of the mutants can have several causes beyond their potential involvement in proton coupling...

>The sentence "Under electrogenic uptake, which indicates coupled transport, and counterflow uptake, which indicates uncoupled transport {Kaback, 2001}, ":

Please reformulate, the current phrasing suggests that protons are not involved in the counterflow transport mode, while they are. They just do not drive the direction of transport. The sentence should read: "electrogenic uptake, which allows substrate accumulation, ..counterflow uptake, which allows substrate equilibration"

Done.

Concerning my minor remark "P2, for high affinity transport couple.. there is no correlation between the coupling ion and .." and the authors comment:

>The reference indicated at the end of this statement (17) refers to a different paper (Pfeiffer et al. 1999), please correct. The effect is interesting. It may also reflect cooperativity in the Na- and amino acid binding sites.

Reference has been corrected.

Concerning minor remark "Fig3A, it is unclear whether accumulation levels or initial rates are depicted. – accumulation levels were used and have been specified in the materials and methods. "

>This should be specified in the legend as well.

Done.

Concerning minor remark "Fig3C/d, Fig 4ACE, specify transport mode and whether initial rates or accumulation levels were used. – accumulation levels were used and have been specified in the materials and methods. "

>This should be specified in the legend as well.

Done.